# Intraclass Correlation in Paired Associative Stimulation and Metaplasticity

**Giuditta Schapira [1], Justin Chang [1], Yeun Kim [1], Jacqueline P. Ngo [1], Choi Deblieck [2], Valentina Bianco [3], Dylan J. Edwards [4], Bruce H. Dobkin [5], Allan D. Wu [6] and Marco Iacoboni [1,7,*]**

1.  Ahmanson-Lovelace Brain Mapping Center, Department of Psychiatry and Biobehavioral Sciences, David Geffen School of Medicine at UCLA, Los Angeles, CA 90095, USA
2.  Antwerp Management School, University of Antwerp, 2000 Antwerpen, Belgium
3.  Department of Movement, Human and Health Sciences, University of Rome "Foro Italico", 00185 Rome, Italy
4.  Moss Rehabilitation Research Institute, Elkins Park, PA 19027, USA
5.  Department of Neurology, David Geffen School of Medicine at UCLA, Los Angeles, CA 90095, USA
6.  Department of Neurology, Feinberg School of Medicine, Northwestern University, Evanston, IL 60208, USA
7.  Brain Research Institute, UCLA, Los Angeles, CA 90095, USA
*   Correspondence: iacoboni@ucla.edu

**Abstract:** Paired associative stimulation (PAS) is a widely used noninvasive brain stimulation protocol to assess neural plasticity. Its reproducibility, however, has been rarely tested and with mixed results. With two consecutive studies, we aimed to provide further tests and a more systematic assessment of PAS reproducibility. We measured intraclass correlation coefficients (ICCs)—a widely used tool to assess whether groups of measurements resemble each other—in two PAS studies on healthy volunteers. The first study included five PAS sessions recording 10 MEPS every 10 min for an hour post-PAS. The second study included two PAS sessions recording 50 MEPS at 20 and 50 min post-PAS, based on analyses from the first study. In both studies PAS sessions were spaced one week apart. Within sessions ICC was fair to excellent for both studies, yet between sessions ICC was poor for both studies. We suggest that long term meta-plasticity effects (longer than one week) may interfere with between sessions reproducibility.

**Keywords:** spike-timing-dependent plasticity; transcranial magnetic stimulation; reliability; long-term potentiation





## 1. Introduction

Paired associative stimulation (PAS) is a noninvasive brain stimulation (NIBS) protocol to assess neural plasticity in neurological patients and healthy controls [1–5]. PAS is inspired by spike timing dependent plasticity (STDP) basic neuroscience protocols.

In STDP, consistently activating the postsynaptic neuron after the presynaptic one generates long-term potentiation (LTP), while consistently activating the presynaptic neuron after the postsynaptic one generates long-term depression (LTD) [6].

PAS can mimic LTP or LTD as an in vivo and non-invasive paradigm to study neural plasticity in humans by coupling an electrical peripheral nerve stimulation with a central transcranial magnetic stimulation (TMS) pulse to induce neuroplastic changes in the motor cortex [1,7–12]. The effects of PAS on the primary motor cortex (M1) are indexed by the changes in motor cortical excitability that can be measured with motor evoked potentials (MEPs) [13] in response to TMS stimulation of M1. Indeed, PAS can either be excitatory or inhibitory, depending on the interstimulus interval (ISI) between electrical peripheral nerve stimulation and central TMS pulse. If the afferent stimulus from the peripheral nerve stimulation reaches the cortex before the delivery of the TMS pulse, it will result in an increase in excitability [14]. On the other hand, if the TMS pulse is delivered before the peripheral stimulation reaches the cortex, the protocol will cause a decrease in excitability.

Electrical stimulation of peripheral nerves in the forearm reaches the somatosensory cortex with a delay of about 20 ms. For this reason, commonly used ISIs, such as 25 ms (PAS-25) or even longer, will have an excitatory effect, while shorter ISIs such as 10 ms (PAS-10) or less will have an inhibitory effect [13]. An increase in the excitability of the motor cortex will result in an increase in the size of MEPs, while a decrease in excitability will show a decrease in the size of MEPs.

Other NIBS have local excitatory and inhibitory effects, akin to PAS. One of the most effective is theta burst stimulation (TBS), a patterned TMS stimulation protocol which can produced prolonged inhibition for about an hour in only 40 s (continuous TBS, or cTBS), and prolonged local excitation in a little more than three minutes (intermittent TBS, or iTBS) [15]. More traditionally, repetitive TMS (rTMS), has also shown rather consistent findings of excitation and inhibition, in its different forms of high frequency rTMS (>1 Hz) and low frequency rTMS (1 Hz or less) [16]. Note that also weak direct current can produce cortical excitation and inhibition with anodal and cathodal stimulation, respectively, yet the nature of the induce plasticity appears to differ, focal in the case of TMS, and non-focal in the case of transcranial direct current stimulation (tDCS) [17,18].

Most of the vast PAS literature so far has focused on excitatory LTP-like protocols, most commonly PAS-25 [7,19–23]. Indeed, in a literature review that we performed first in early 2020, with subsequent updates in late 2020 and Spring 2022 (see details in the Methods), we retrieved almost 400 published articles. Of these, 150 used the traditional paradigm in healthy volunteers, as in the present study (Study 1 in particular), while many others adopted variations of it. PAS has been applied in both health (n = 150) and disease (n = 148) and coupled with pharmacological interventions to modulate plasticity (n = 17). Among the 150 studies applying PAS to healthy participants, 65% used LTP-like protocols, 32% used both LTP- and LTD-like protocols, and only 3% used LTD-like protocols only. The preponderance of LTP-like protocols in the literature led to also adopt an LTP-like PAS protocol in our study.

To the best of our knowledge, only two studies assessed PAS reproducibility in healthy volunteers. To do so, both studies used the Intraclass Correlation Coefficient (ICC), a widely used statistical tool to assess whether groups of measurements resemble each other. Indeed, ICC is so widely used and discussed in the literature that Pubmed queries in October 2022 returned 33,172 results for Intraclass Correlation Coefficient and 38,152 results for Intraclass Correlation.

Disappointingly, the two studies using ICC on PAS reached rather different conclusions. A first study suggested that PAS reproducibility is very low [21], whereas a second study suggested that PAS reproducibility increases if assessed in the afternoon [22]. Both studies assessed PAS effects on motor excitability almost immediately after PAS. Indeed, one study assessed PAS effects right after PAS [21], whereas the second study assessed it five minutes after PAS [22]. Given that most studies assessing PAS effects make measurements at multiple timepoints for about an hour after PAS, and that PAS effects on motor excitability tend to show an increase over time after PAS, the two previous studies on PAS reliability may have used a too narrow temporal window to thoroughly assess PAS reliability. Furthermore, ICC is subject to several conditions to be met to provide unbiased results, as discussed below in relation to our own study and in a recent meta-analysis on TMS outcomes that includes several TMS studies (not using PAS though) using ICC [24]. Both previous papers applying ICC to PAS do not report whether those conditions were met in their datasets.

Therefore, to revisit this issue, we designed and performed a first study using a widely adopted LTP-like protocol, testing PAS effects every ten minutes for an hour post-PAS. On this dataset, we performed analyses on a progressively increasing number of MEPs to establish a minimum number of MEPs that provides a stable response at individual level. We performed permutation tests on this dataset to simulate a normal distribution under the null hypothesis of no significant difference in MEPs relative to baseline, to assess the optimal number of datapoints (MEPs) per timepoint to obtain more reliable PAS effects [25].

We subsequently performed a second study based on these analyses. Finally, we computed and compared ICCs in both studies.

## 2. Materials and Methods

### 2.1. Literature Review and Study Design

Since we were interested in assessing PAS reliability, we thought of repeatedly testing our study participants. However, practical considerations make it difficult to deploy such a design on a large sample. Therefore, we reviewed the existing literature to obtain quantitative information on the number of post-PAS MEPs per study as well as per participant had been measured in previously published studies. This information allowed us to design the two studies in such a way that they fell within the top half (total MEPs per study) and top quartile (total MEPs per participant) of the literature, assuring that our dataset compares very well with previously published studies on these parameters (Figure 1).

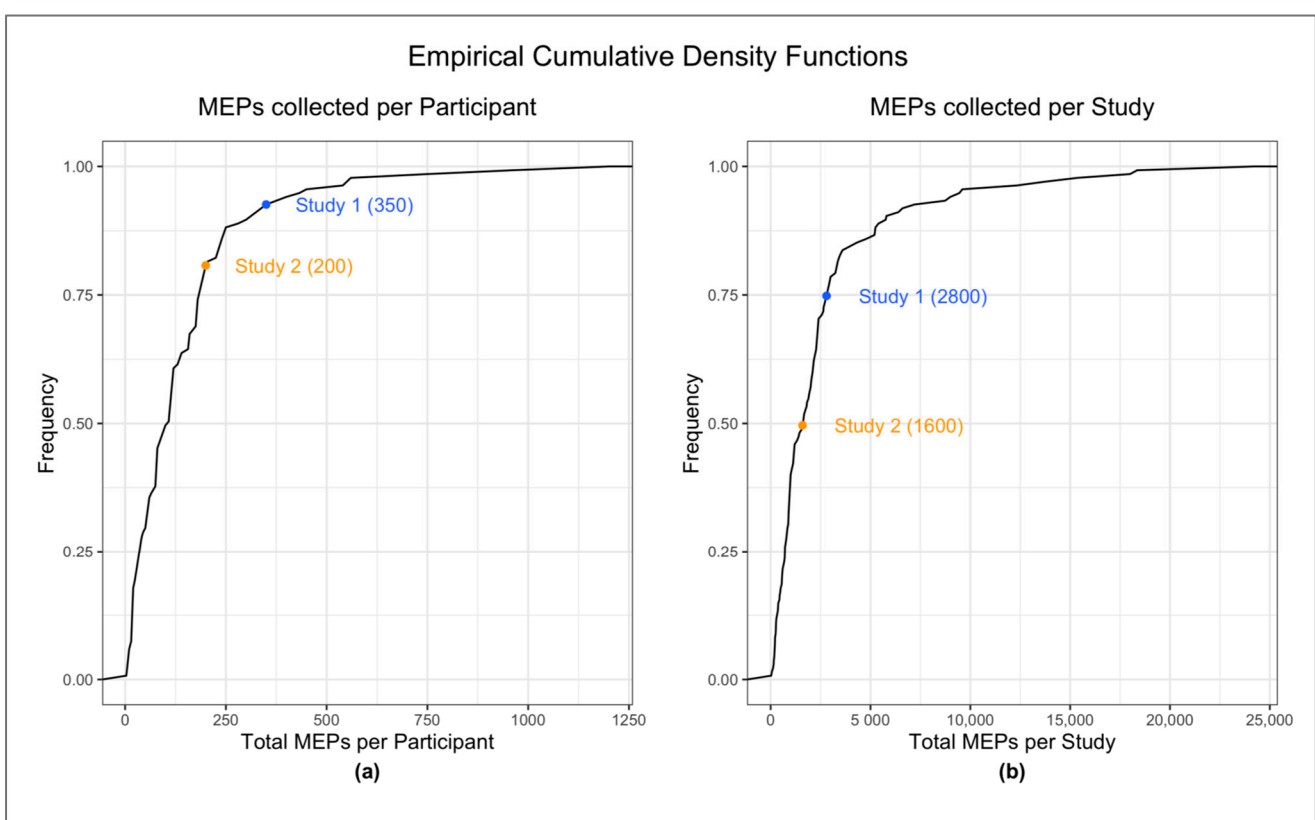

**Figure 1.** Empirical cumulative density functions of (**a**) total post-PAS MEPs collected per participant and (**b**) total post-PAS MEPs collected per study. Blue and yellow dots indicate where Studies 1 and 2 fall, respectively, compared to previously published studies. None of the values in the figure include MEPs collected at baseline.

### 2.2. Participants

Ten healthy participants were recruited for the first study (Study 1) and eleven healthy participants for the second study (Study 2). In Study 1, healthy volunteers participated in five PAS sessions spaced one week apart [25]. In Study 2 healthy volunteers participated in two PAS sessions, also spaced one week apart. Since one of the main motivations of the two studies is to test whether PAS can be used to assess individual levels of plasticity that can be used to individualize treatments in patients, participants were not studied at specific times of the day. Indeed, such a constraint would be impractical for clinical studies. The participants were all right-handed, with no history of alcohol or drug abuse, and no prior diagnosis of any neurological or psychiatric disorders [25]. The participants

reported not taking any psychoactive medications in the last month [25]. Two participants were excluded from Study 1 due to technical issues leading to missing baseline MEP data from two different sessions [25]. Three participants were excluded from Study 2 due to missing data. Data analysis was conducted on the remaining eight participants from Study 1 (six females; mean age 24.25 ± 6.18 years) and the remaining eight participants from Study 2 (five females; mean age 23 ± 6.7 years).

### 2.3. Electromyographic Recording

For both Study 1 and Study 2 participants sat in an armchair for the duration of the experiment. Surface electromyography (sEMG) electrodes (Delsys 2.1, Delsys, Inc., Natick, MA, USA) were attached to the right first dorsal interosseus (FDI) muscle to measure MEPs. The electrical activity of the muscle was amplified, and bandwidth filtered between 20 Hz and 450 Hz (Bagnoli EMG, Delsys, Inc.), digitized with a sample rate of 5 kHz (CED 1401 running Signal V4 software, Cambridge Electronic Design, Cambridge, UK), and stored for offline analysis using custom Matlab routines [25].

### 2.4. Transcranial Magnetic Stimulation

#### 2.4.1. "Hotspot" Identification and TMS Delivery

Transcranial magnetic stimulation (TMS) was delivered through a figure-eight coil (70 mm diameter each coil) connected to a Magstim Bistim2. TMS pulses were delivered over the left M1 of each participant to assess pre- and post-PAS effects. In Study 1, TMS pulses were also delivered to the right M1 to serve as control [25]. However, these data, being control data, were not used for ICC measurements in this study. In Study 1, the coil was held so that the handle pointed posteriorly at a 45-degree angle down the sagittal plane and moved systematically across the participant's scalp to identify the "hotspot". The "hotspot" was defined as the point where TMS consistently produced the largest MEPs from the contralateral FDI. The location of the "hotspot" was recorded onto a stereotactic system (Brainsight, Rogue Research, Montreal, Canada) to ensure that the coil would be held in the optimal position for the duration of the study [25].

In Study 2, the coil was held exactly as in Study 1, but the hotspot was determined using 3D coordinate locations in an MNI 152 template aggregated by adopting methods outlined by Mayka et al. (2006), which we think make hotspot determination more easily reproducible. The authors used fMRI and PET data from 278 individuals across 68 studies to correlate coordinate locations with specific areas of the brain [26]. We used a 95% confidence interval of the estimated M1 as center coordinate location. We also added a grid pattern in a diamond formation with four additional 'targets' spaced 10 mm × 10 mm apart. The center spacing used for Left M1 was x = −37 mm, y = −21 mm, and z = 58 mm with a volume size of 5 mm spherical in 3D. All targets were projected to the skin surface level of the MNI model in Brainsight to relay an accurate coordinate location at the surface of the scalp that would correspond to the brain area we were trying to target. At the beginning of each session, five target areas were tested to find a location where 3 out of 6 MEPs were larger than 0.5 mV. The starting maximum stimulator output (MSO) was 56%, and the intensity was individually tailored to obtain consistent MEPs for each participant. The point that consistently showed the largest MEPs was chosen as the "hotspot" and recorded onto Brainsight. Visual feedback from Brainsight ensured that coil positioning was kept at the optimal location during the entire session.

#### 2.4.2. Resting Motor Threshold

In Study 1, the resting motor threshold (rMT) for the chosen "hotspot" was determined as the lowest stimulus intensity required to elicit MEPs of 0.05 mV or higher in at least 3 out of 6 consecutive trials [25]. In Study 2 we used simple adaptive PEST (SA-PEST) (Adaptive PEST TMS motor threshold assessment tool, Brain stimulation laboratory, Department of Psychiatry, Medical University of South Carolina) with the following values: initial output value obtained from the previous step, step size of 6% MSO. SA-PEST is a nonparametric

protocol that involves the application of simple decision rules and uses the history of successes/failures to determine stimulus intensities to try [27,28]. A certain MSO is tested, and an MEP is elicited. If the resulting MEP > 0.5 mV the software receives a "yes" input. If the resulting MEP < 0.5 mV the software receives a "no" signal [29,30]. Depending on the yes/no input the software generates a new MSO to test until a 50% balance between yes/no is reached.

### 2.4.3. Baseline Corticospinal Excitability

In Study 1, baseline corticospinal excitability (CSE) was assessed by delivering 12 TMS pulses at a random interstimulus interval between 8–12 s at an intensity that would consistently produce MEPs in the 0.5 to 0.75 mV range in the relaxed FDI [25]. The first two MEPs were automatically excluded. In Study 2, a stimulus intensity of 110% of the individual rMT was used to measure baseline CSE. TMS pulses were given at a random interstimulus interval between 8–12 s to obtain 25 baseline MEPs. In both studies MEP collection only acquired MEPs with a latency of no less than 20 ms from stimulation, a minimum peak-to-peak distance of 0.05 mV, and no presence of voluntary movement during the pre-stimulus phase. To control for attention, subjects were asked to look at the target muscle and count the number of twitches [31].

### 2.5. Paired Associative Stimulation

PAS was performed in both studies in every session immediately after baseline CSE. It consisted of repetitive single electric stimulation (square wave, 0.2 ms duration) to the right ulnar nerve delivered using a DS3 Isolated Current Stimulator (Digitimer North America, Ft. Lauderdale, FL, USA) followed by a single TMS pulse delivered over the left motor cortex with an ISI of 25 ms [25,32]. The ulnar nerve was stimulated at an intensity of 300% perceptual threshold [11] and TMS intensity was set at 130% rMT [32]. In total, 200 pairs of stimulation were applied at 0.25 Hz.

### 2.6. Post-PAS Assessment

In both studies, TMS pulses were delivered with the same intensity used for baseline MEP assessment and collected with a random ISI between 8 and 12 s. In study 1, MEPs were collected immediately after PAS and in 10 min intervals for a period of 60 min [25]. At each timepoint, 12 MEPs were collected, and the first two MEPs were always discarded [25]. In Study 2 TMS pulses were delivered 20 min and 50 min after PAS. At each timepoint, 50 MEPs were collected, and no MEP was discarded.

### 2.7. Measures

The collected data consisted of MEPs—waveforms of electromyographic activity in response to the excitation of the motor cortex from TMS. In Study 1, in five separate sessions, over five consecutive weeks, 10 MEPs were collected for final analysis at baseline and then every 10 min for a period of 60 min for a total of 400 MEPs per participant [25]. In Study 2, in two separate sessions, over two consecutive weeks, 25 MEPs were collected at baseline, and 50 MEPs were collected at 20 min and at 50 min after PAS, for a total of 250 MEPs per participant over the two sessions. The decision to collect 50 MEPs was based on the analyses performed on Study 1 data, which suggested that 50 MEPs provide more reproducible results [10]. Given that collecting 50 MEPs per timepoint requires more time than collecting only 10 MEPs, we decided to have 30 min between the two timepoints. Furthermore, our previous analyses of Study 1 data [10] suggested that more reproducible results are obtained at timepoints from 20 min after PAS. Hence, we planned our first timepoint for Study 2 at 20 min after PAS.

The data were corrected for direct current offset to ensure that the EMG signal within the pre-stimulus phase had an average of 0 mV [25]. The window for the calculation of the MEP waveform began at 20 ms after the stimulus onset and lasted for 50 ms. Thereafter, the area-under-the-curve (AUC) was computed by taking the integral using the trapezoidal rule after the data were rectified. The digital signals were not further smoothed or filtered to preserve the integrity of the waveforms.

### 2.8. Literature Review Details

We conducted a review of the literature available on PAS to test whether Studies 1 and 2 are comparable with the existing literature. The review was carried out on PubMed on 20 January 2020, and updated on 27 September 2020 and on 26 April 2022. We searched for "Paired Associative Stimulation" in the Title/Abstract. We examined the available studies on healthy subjects exploring peripheral motor PAS simulation. We excluded duplicates, review articles, and papers not available in English. A total of 150 papers were included in the review. Based on the literature review findings we concluded that, compared to the existing literature, the present study provides sufficient MEP data (from Study 1 as well as from Study 2) to draw valid conclusions about the reliability of PAS. Indeed, as shown by Figure 1 above, the two studies fall within the top half (total MEPs per study) and top quartile (total MEPs per participant) of the literature.

### 2.9. Statistical Analyses

#### 2.9.1. Effects of PAS25 on Overall Motor Excitability

In both Study 1 and Study 2, for each timepoint in each participant, we computed the median MEP and excluded MEPs above two standard deviations from the medians as outliers. This procedure resulted in removing 130 (4.1%) post-PAS MEPs in Study 1 and 106 (5.3%) in Study 2. A paired *t*-test was conducted to compare average MEP size before and after PAS25 intervention. *t*-test analysis was conducted in SPSS version 26 [33].

#### 2.9.2. Temporal Effects of PAS25 on Motor Excitability

We used each participant's median baseline MEP as the normalization factor for their post-PAS MEPs. A Kolgomorov–Smirnov test determined that the data was not normally distributed [34]. Therefore, Box-Cox transformation was used to fit the data for normality [25,35]. A repeated measures ANOVA with session and timepoint as within-subjects variables was used to test for significant timepoint, session, and timepoint by session interaction effects. Both Kolgomorov–Smirnov test and Box–Cox transformations were performed on SPSS version 26 [33].

#### 2.9.3. Intraclass Correlation

Intraindividual reliability of post-PAS MEPs was assessed using intraclass correlation coefficients (ICC). ICC estimates of consistency and their 95% confidence intervals were calculated using SPSS statistical package version 26 based on sessions (Study 1: k = 5, Study 2: k = 2), 2-way random-effects model [33,36,37]. We calculated ICCs between sessions, across timepoints within session, and across sessions for each timepoint. Table 1 illustrates the different combinations of data used to obtain ICCs in Study 1 and Study 2. We report ICC estimates of consistency and their confidence interval rather than *p* values since the former are considered more meaningful than the latter [19]. According to guidelines proposed in [19], based on the 95% confidence interval of the ICC estimate, values less than 0.5, between 0.5 and 0.75, between 0.75 and 0.9, and greater than 0.90 are considered of poor, moderate, fair, and excellent reliability, respectively.

While ICC analysis is traditionally used to assess test–retest, intra-rater, and inter-rater reliability, it is subject to a number of conditions that must be met to provide unbiased results, including assumptions of homogeneity of variance and normality [38–40]. Because these conditions are rarely met in practice, ICC analysis can often give biased results [41]. For this reason, we opted to validate the results obtained through ICC analysis through

Lin's concordance correlation coefficient (CCC) [42,43]. Like ICC, CCC is a reliability and agreement index. While ICC results may depend on certain assumptions, including normality, CCC results do not [41]. However, unlike ICC, which can be run on any number of measurements, CCC can only be applied to pairs of measurements [43]. Therefore, we took a different approach to running CCC on Study 1 by grouping the data. To run CCC across timepoints within each session we divided the data into two groups: $G_1$ (10, 20, 30 min) and $G_2$ (40, 50, 60 min) (see Table 2). We excluded from CCC analysis MEPs collected immediately post-PAS (0 min). Such exclusion was based on the lack of stability of MEPs collected immediately following PAS intervention. Firstly, permutation tests conducted by Kim et al., showed that MEPs collected immediately after PAS are the least stable [25]. Secondly, Fratello et al., tried and failed to demonstrate PAS reliability by looking solely at MEPs collected immediately following PAS intervention [21,25]. To run CCC across sessions for each timepoint we ran pairwise analyses. Table 2 illustrates the different combinations of data used to obtain CCCs in Study 1 and Study 2. CCC analysis was conducted in R (v1.2.5042).

**Table 1.** Illustration of the combinations used to calculate ICCs.

| | | Combination * | ICCs Obtained |
|---|---|---|---|
| Between sessions | Study 1 | S1 vs. S2 vs. S3 vs. S4 vs. S5 | 1 |
| | Study 2 | S1 vs. S2 | 1 |
| Across timepoints for each session | Study 1 | S1 (0 min vs. 10 min vs. 20 min vs. 30 min vs. 40 min vs. 50 min vs. 60 min), . . . S5 (0 min vs. 10 min vs. 20 min vs. 30 min vs. 40 min vs. 50 min vs. 60 min) | 5 |
| | Study 2 | S1 (20 min vs. 50 min), S2 (20 min vs. 50 min) | 2 |
| Across sessions for each timepoint | Study 1 | 0 min (S1 vs. S2 vs. S3 vs. S4 vs. S5), . . . 60 min (S1 vs. S2 vs. S3 vs. S4 vs. S5). | 7 |
| | Study 2 | 20 min (S1 vs. S2), 50 min (S1 vs. S2) | 2 |

* S stands for session.

**Table 2.** Illustration of the combinations used to calculate comparable ICCs and CCCs for validation of consistency.

| | | Combination | CCC Obtained |
|---|---|---|---|
| Between sessions | Study 1 | S1 vs. S2, S1 vs. S3, S1 vs. S4, S1 vs. S5, S2 vs. S3 . . . | 10 |
| | Study 2 | S1 vs. S2 | 1 |
| Across timepoints within each session | Study 1 | S1 ($G_1$ vs. $G_2$), S2 ($G_1$ vs. $G_2$), . . . S5 (G1 vs. $G_2$) | 5 |
| | Study 2 | S1 (20 min vs. 50 min), S2 (20 min vs. 50 min) | 2 |
| Across sessions for each timepoint | Study 2 | 20 min (S1 vs. S2), 50 min (S1 vs. S2) | 2 |

## 3. Results

### 3.1. Effects of PAS on Motor Excitability

In Study 1, there was a significant difference in MEP size from before PAS25 intervention (M = 2.86, SD = 1.17) to after (M = 5.30, SD = 2.37); t (7) = −3.02, *p* = 0.020).

In Study 2, however, there was no significant difference in MEP size from before PAS25 intervention (M = 1.83, SD = 1.35) to after (M = 2.34, SD = 1.68); t (7) = −1.42, *p* = 0.200).

### 3.2. Temporal Effects of PAS25 on Motor Excitability

In Study 1, repeated measures ANOVA revealed an effect of timepoint (F(6, 42) = 6.884, *p* < 0.0001, ηp2 = 0.496), showing an increase over time of post-PAS MEPs fitting a linear trend (LINEAR CONTRAST: F(1, 7) = 17.143, *p* = 0.004, ηp2 = 0.710). No effect of session (F(4, 28) = 1.583, *p* = 0.206, ηp2 = 0.184) or interaction between timepoint and session (F(24, 168) = 1.235, *p* = 0.219, ηp2 = 0.150) was observed (Figure 2).

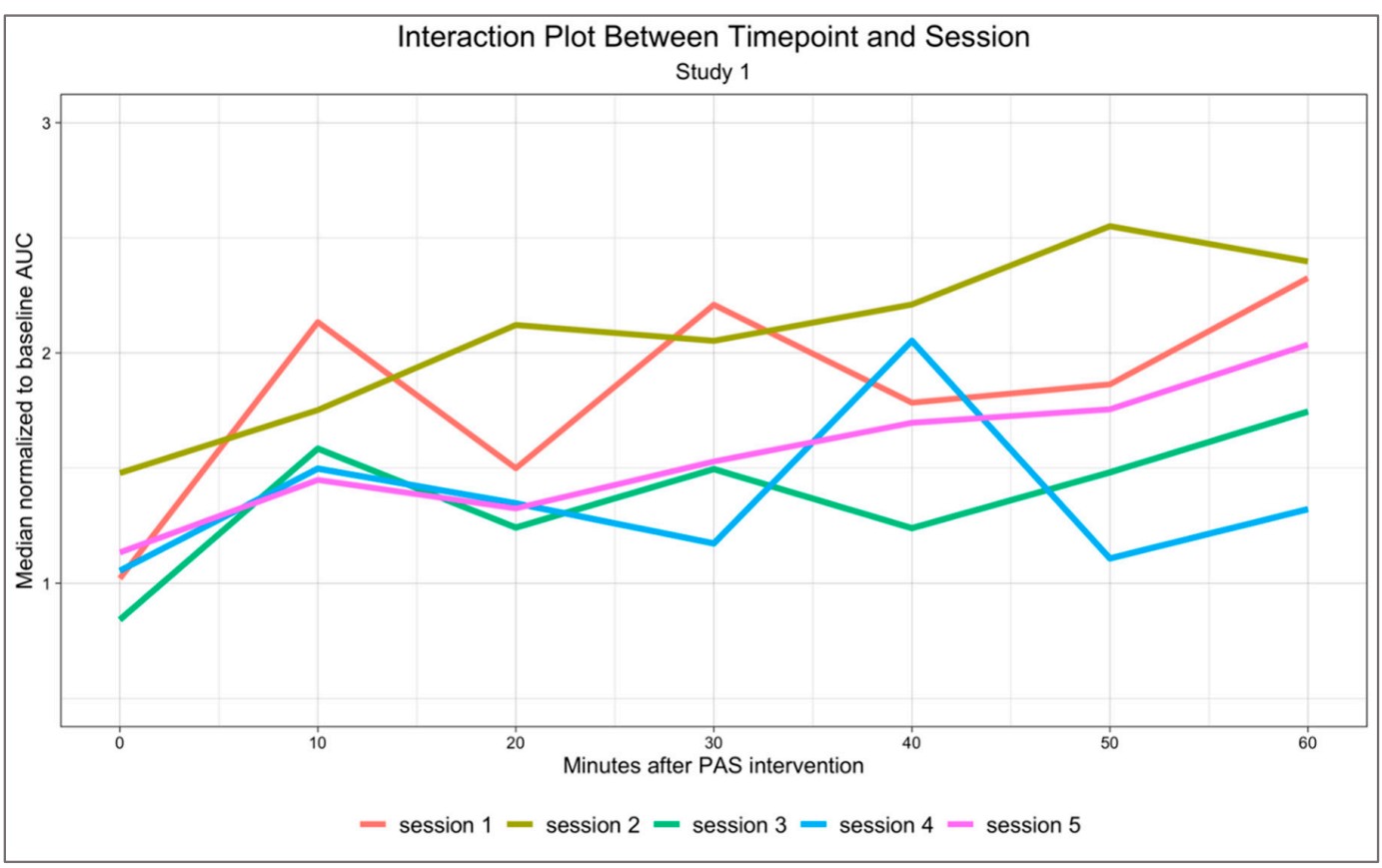

**Figure 2.** Interaction plot of Study 1. Significant effect of timepoint (F = 6.884, *p* < 0.0001). No significant session (F(4, 28) = 1.583, *p* = 0.206, ηp2 = 0.184) or timepoint by session (F(24, 168) = 1.235, *p* = 0.219, ηp2 = 0.150) interaction.

Study 2, however, did not show any effect of timepoint (F(1, 7) = 0.691, *p* = 0.433, ηp2 = 0.090), session (F(1, 7) = 1.134, *p* = 0.322, ηp2 = 0.139), or interaction between timepoint and session (F(1, 7) = 244, *p* = 0.636, ηp2 = 0.034) (Figure 3).

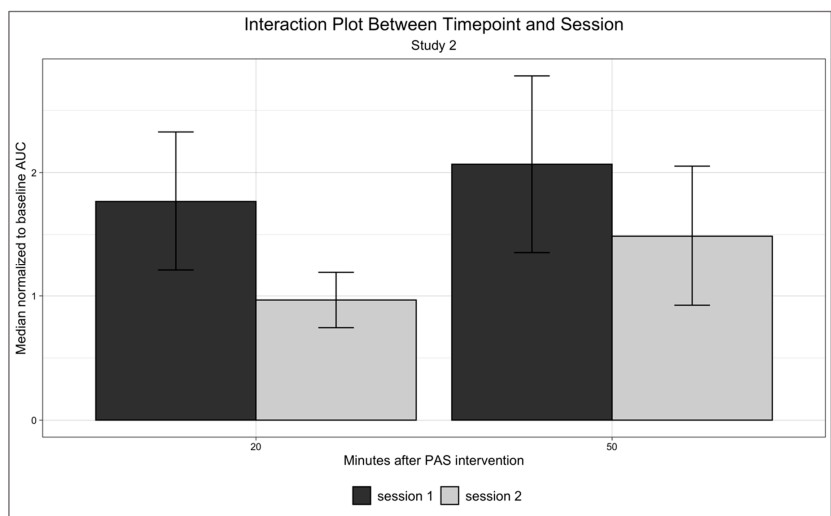

**Figure 3.** Interaction plot of Study 2. No significant effect of timepoint (F(1, 7) = 0.691, *p* = 0.433, ηp2 = 0.090), session (F(1, 7) = 1.134, *p* = 0.322, ηp2 = 0.139), or interaction between timepoint and session (F(1, 7) = 244, *p* = 0.636, ηp2 = 0.034). Since only two timepoints were used in Study 2, we opted to graph the results with bar graphs here. Error bars represent standard errors.

### 3.3. ICC-CCC Comparisons and Remaining ICC Results

ICCs and CCCs used in the regression analysis for Study 1 and Study 2 are shown in Tables 3 and 4, respectively. Comparison of CCC values to their corresponding ICC values confirmed the validity of ICC. Simple linear regression analysis was performed to test if CCC significantly predicted ICC score. The analysis was performed using SPSS statistical package version 26 [33]. For Study 1, the regression was statistically significant ($R^2$ = 0.85, F(1, 13) = 76.42, *p* < 0.0001). For Study 2, the regression was also statistically significant ($R^2$ = 0.97, F(1, 3) = 103.8, *p* = 0.002). CCC significantly predicted ICC in both Study 1 (β = 0.5606, *p* < 0.0001) and Study 2 (β = 1.3148, *p* = 0.002) (Figure 4).

The ICC and CCC values for both Study 1 and Study 2 are generally fair to excellent across timepoints within each session (with the exception of Session 5 of Study 1) but generally poor to fair between sessions or across sessions for each timepoint. Table 5 shows the remaining ICCs from Study 1 that have no corresponding CCCs.

**Table 3.** ICCs and CCCs used in the regression analysis for Study 1. In parenthesis the 95% confidence interval. $G_1$ corresponds to the median MEP at 10, 20, 30 min and $G_2$ corresponds to the median MEP at 40, 50, 60 min.

|  |  | ICC (95% CI) | CCC (95% CI) |
|---|---|---|---|
| Across timepoints within each session ($G_1$ vs. $G_2$) | S1 | 0.84 (0.21–0.97) | 0.71 (0.12–0.93) |
|  | S2 | 0.96 (0.79–0.99) | 0.81 (0.49–0.94) |
|  | S3 | 0.58 (−1.12–0.92) | 0.48 (0.35–0.84) |
|  | S4 | 0.97 (0.85–1.00) | 0.92 (0.67–0.98) |
|  | S5 | 0.90 (0.52–0.98) | 0.55 (0.09–0.82) |
| Between sessions | S1 vs. S2 | 0.051 (−3.74–0.81) | 0.02 (−0.5–0.53) |
|  | S1 vs. S3 | 0.36 (−2.18–0.871) | 0.2 (−0.47–0.73) |
|  | S1 vs. S4 | 0.57 (−1.14–0.91) | 0.38 (−0.19–0.76) |
|  | S1 vs. S5 | 0.60 (−0.99–0.92) | 0.43 (−0.08–0.76) |
|  | S2 vs. S3 | 0.42 (−1.90–0.88) | 0.15 (−0.25–0.5) |
|  | S2 vs. S4 | 0.76 (−0.20–0.95) | 0.59 (0.01–0.87) |
|  | S2 vs. S5 | −0.48 (−6.40–0.70) | −0.13 (−0.45–0.21) |
|  | S3 vs. S4 | −0.12 (−4.60–0.78) | −0.05 (0.47–0.39) |
|  | S3 vs. S5 | −1.16 (−9.80–0.57) | −0.31 (−0.72–0.28) |
|  | S4 vs. S5 | 0.15 (−3.23–0.83) | 0.08 (−0.23–0.38) |

**Table 4.** ICCs and CCCs used in the regression analysis for Study 2. In parenthesis is the 95% confidence interval.

|  |  | ICC (95% CI) | CCC (95% CI) |
|---|---|---|---|
| Across timepoints within each session | S1 | 0.91 (0.53–0.98) | 0.83 (0.37–0.96) |
|  | S2 | 0.86 (0.27–0.97) | 0.72 (0.32–0.9) |
| Across sessions for each timepoint | 20 min | 0.65 (−0.73–0.93) | 0.42 (−0.14–0.74) |
|  | 50 min | 0.52 (−1.38–0.90) | 0.34 (−0.39–0.81) |
| Between sessions | S1 vs. S2 | 0.62 (−0.91–0.92) | 0.41 (−0.14–0.78) |

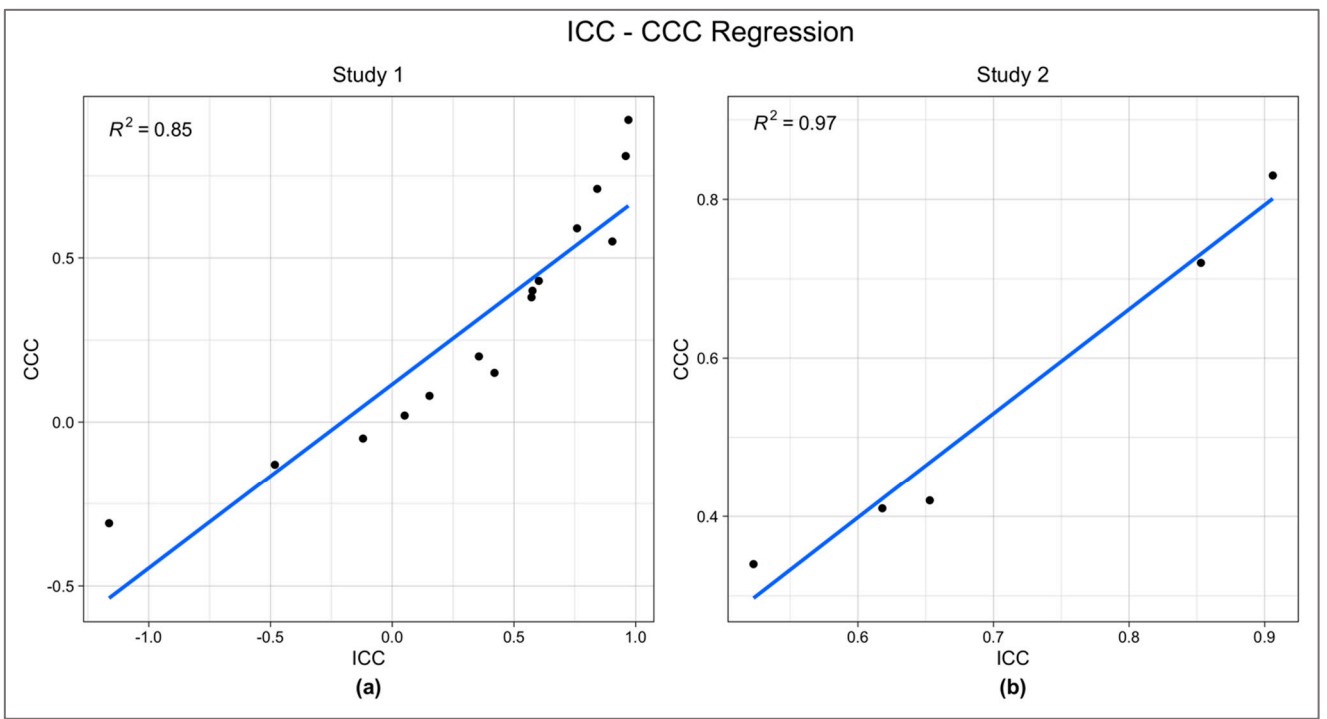

**Figure 4.** Simple linear regression analysis testing the correlation between ICC and CCC in Study 1 (**a**) and Study 2 (**b**). Both studies showed significant correlations (Study 1: $R^2$ = 0.97, $F(1, 3)$ = 103.8, $p$ = 0.002), (Study 2: $R^2$ = 0.97, $F(1, 3)$ = 103.8, $p$ = 0.002). In both studies CCC significantly predicted ICC (Study 1: β = 0.5606, $p$ < 0.0001), (Study 2: β = 1.3148, $p$ = 0.002).

**Table 5.** Remaining ICCs from Study 1. In parenthesis the 95% confidence interval.

|  |  | ICC (95% CI) |
|---|---|---|
| Across timepoints within each session | S1 | 0.86 (0.63–0.97) |
|  | S2 | 0.96 (0.88–0.99) |
|  | S3 | 0.84 (0.57–0.96) |
|  | S4 | 0.97 (0.93–0.99) |
|  | S5 | 0.70 (0.21–0.93) |
| Across sessions for each timepoint | 0 min | 0.22 (−1.18–0.82) |
|  | 10 min | 0.33 (−0.87–0.85) |
|  | 20 min | 0.45 (−0.54–0.87) |
|  | 30 min | 0.72 (0.21–0.94) |
|  | 40 min | 0.04 (−1.66–0.78) |
|  | 50 min | 0.66 (0.05–0.92) |
|  | 60 min | 0.67 (0.07–0.90) |
| Between sessions | S1 vs. S2 vs. S3 vs. S4 vs. S5 | 0.58 (−0.18–0.90) |

## 4. Discussion

Paired associative stimulation (PAS) is a successful noninvasive brain stimulation protocol that mimics at systems level spike timing dependent plasticity (STDP) and its effects. In the two decades since the publication of the first paper proposing PAS [11], a flourishing literature has adopted the paradigm in health and disease. Only two studies, however, tested its reliability, with less than exciting results. One study found basically no reliability at all [21], while the other proposed that better reliability could be achieved by performing PAS in the afternoon [22]. Both studies, however, tested for PAS effects only shortly after PAS (whereas PAS effects are typically tracked also at later times), and did not report whether conditions for unbiased ICC results were met. Since PAS is potentially a powerful noninvasive stimulation protocol that could be used to assess individual level plasticity and its changes after interventions, we designed two studies to further investigate this issue. For Study 1, we adopted a widely used protocol that assesses PAS effects every ten minutes for an hour after PAS. We subsequently used data from Study 1 to estimate whether more MEPs may provide more stable estimates of PAS effects and then designed Study 2 based on those results. In Study 2, we assessed PAS effects at two timepoints only (20 and 50 min after PAS) but collected 50 MEPs per timepoint.

We then used CCC, whose results are not affected by various aspects of the data, including normality of distribution, to validate our ICC results, an approach that the previous two studies did not take. CCC and ICC values were highly correlated, suggesting that ICC values were unbiased. Based on these analyses, we also assume that ICC results that could not be validated by CCC comparisons, are unlikely biased.

The overall pattern of these analyses shows that across timepoints within session ICCs and CCCs are generally fair to excellent for both Study 1 and 2. Between sessions ICCs and CCCs, however, are poor to fair for both Study 1 and 2, even though the two studies show differences between them, with PAS effects and temporal effects on motor excitability in Study 1 but not in Study 2.

Therefore, this study obtains similar results to those reported by the previous studies of Fratello et al. and Sale et al. [21,22], even though it extended the analysis of between sessions PAS reliability at much later timepoints than the previous two studies.

How to explain the difference between within session ICCs and CCCs values and between sessions ICCs and CCCs values? One potential explanation may be due to the many factors that affect neural plasticity and may influence PAS effects from session to session [17,22,25,31,44–53]. While those factors could be controlled in the lab, they are not realistically controlled in clinical studies, which is the potential real-world application of PAS. Indeed, PAS could potentially be used to track in a reliable way effects of different kinds of interventions—pharmacological, behavioral, or brain stimulation—on neural plasticity. However, to do so, PAS between sessions reliability is required. Therefore, it is crucial to also provide an alternative explanation to the current and previous findings and possibly test them in future studies [21,22].

Our study, as in Fratello et al. and Sale et al. used a one-week temporal interval between sessions. It is generally believed that spacing two PAS sessions one week apart should be long enough to prevent carry over effects from the previous session that may potentially generate metaplasticity effects in the subsequent session, which would make it largely impossible to reproduce the same pattern of PAS effects from one session to another one [54–57]. However, this widespread belief may not be strongly supported by empirical data. Metaplasticity is a well-documented phenomenon according to which the history of prior activity modulates synaptic activity mechanisms. A study in freely moving rats demonstrates that the raised threshold for LTP induction produced by conditioning stimulation recovered rather slowly, over a 7- to 35-day period [58]. These data suggest that between sessions, PAS reproducibility should be tested using much longer temporal intervals to avoid potential metaplasticity effects. These longer temporal intervals may also better fit potential real world applications of PAS to assess the effects of interven-

tions or treatments on neural plasticity in clinical studies, which generally unfold over several weeks.

An important consideration, however, is whether TMS-induced MEPs are themselves reliable. If it turns out that they are not, it is hard to see how PAS itself can be reliable, since MEPs are the most used outcome measure of PAS protocols. A recent meta-analysis on reliability of TMS outcome measures has recently raised concerns about the fact that most studies are not rather conclusive or generalizable regarding MEP reliability [24]. Nevertheless, the within sessions ICC was fair to excellent for both Study 1 and 2, whereas the between sessions ICC was poor for both Study1 and 2. This pattern of results can be hardly explained by a potential low reliability of TMS-induced MEPs.

Another important consideration regarding our study is the sample size. It is obviously small in both Study 1 and Study 2, generating concerns about its generalizability. The small sample size is due to a design choice focused on reproducibility of results that in Study 1 led us to repeat PAS on the same participants in five separate weekly sessions, which we believe is the highest number of PAS sessions performed in a single study. The main motivation behind this design choice, however, is to test whether PAS could be used to assess individual levels of plasticity that can guide personalized TMS treatment in clinical populations. Since personalized treatments always focus on a single individual, generalizability is not a major concern.

To conclude, this study confirms the previous findings of lack of between sessions reproducibility of PAS effects, while testing it over a much longer within-session temporal window. It also proposes that much longer intersession intervals should be tested in future studies, to rule out that between session PAS effects fail to be reproducible due to the broad sensitivity of synaptic plasticity to the history of prior activity [58].

**Author Contributions:** Conceptualization, M.I., A.D.W., D.J.E. and B.H.D.; methodology, M.I. and Y.K.; software, Y.K.; validation, M.I., G.S. and J.C.; formal analysis, G.S. and J.C.; investigation, C.D. and V.B.; resources, M.I.; data curation, C.D., V.B., Y.K. and J.P.N.; writing—original draft preparation, G.S., J.C., V.B. and M.I.; writing—review and editing, M.I. and all co-authors; visualization, G.S. and J.C.; supervision, M.I.; project administration, M.I., C.D., V.B. and A.D.W.; funding acquisition, B.H.D. All authors have read and agreed to the published version of the manuscript.

**Funding:** This research was funded by the Miriam and Sheldon G. Adelson Medical Research Foundation.

**Institutional Review Board Statement:** The study was conducted in accordance with the Declaration of Helsinki and approved by the Institutional Review Board of UCLA (IRB#12-001045, 11/13/2012).

**Informed Consent Statement:** Informed consent was obtained from all subjects involved in the study.

**Data Availability Statement:** Data sharing will be implemented through a public project in Open Science Framework. Shared de-identified data will be as follows: individual electromyographic data, metadata, and associated documentation that facilitate interpretation of the shared data.

**Acknowledgments:** We thank Ian Heimbuch for assistance with data management. For their generous support, the authors also wish to thank the Brain Mapping Medical Research Organization, Brain Mapping Support Foundation, Pierson-Lovelace Foundation, The Ahmanson Foundation, William M. and Linda R. Dietel Philanthropic Fund at the Northern Piedmont Community Foundation, Tamkin Foundation, Jennifer Jones-Simon Foundation, Capital Group Companies Charitable Foundation, Robson Family and Northstar Fund.

**Conflicts of Interest:** The authors declare no conflict of interest. The funders had no role in the design of the study; in the collection, analyses, or interpretation of data; in the writing of the manuscript, or in the decision to publish the results.

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
