# Peer review of "Intraclass Correlation in Paired Associative Stimulation and Metaplasticity"

_neurosci, doi:10.3390/neurosci3040042_

Round 1
Reviewer 1 Report
The authors have conducted a study in which they evaluate the inter- and intra-session CCI of two PAS protocol experiments.
The authors suggest that long term 23 meta-plasticity effects (longer than one week) may interfere with between sessions reproducibility.
I have some minor comments and suggestions for the authors.
Could the authors give in the introduction some reference to the measurement of icc in other studies.
The authors use the Adaptive PEST TMS motor threshold assessment tool to calculate the resting motor threshold, they mark yes when the potential is greater than 0.5mv and not the other way around. Could the authors give some clarification or reference to this methodology?
Reviewer 2 Report
In this neurophysiologic study, the authors aimed to assess the reproducibility of spike-timing-dependent plasticity, induced by a modified paired associative stimulation protocol, as shown by interclass correlation index calculated for both within and between sessions of post-stimulation motor evoked potentials recordings, in two separate experiments performed in healthy volunteers. Results showed that the effects of stimulating paradigm may interfere with between sessions reproducibility.
The study would be of some value since it further expands well-known literature in the field. However, I have several concerns which can potentially preclude the publication in the present form.
First, the novelty of this investigation is rather low. The authors should also focus on the issue of intra-subject and inter-subject variability of MEPs following NIBS paradigms, and how this would affect also the reproducibility of plasticity effects within and between sessions.
The sample size of both experiments here reported is rather small. Also, crucial data of participants are missing. Lastly, the sample of participants prevents the generalization of results.
Concerning the study design, it is unclear to me whether the two experiments largely differ in terms of the experimental paradigm. It is therefore hard to compare data from a different experimental approach. Also, the authors have included only paradigms able to induce long-term potentiation like plasticity. What about LTD? Is it expected a similar behaviour?
The discussion of findings and references to previous works should be expanded. Also, the authors would report differences and similarities between paired associative stimulation and otherwise NIBS protocols in terms of plasticity effects.
Figures and tables should be improved.
Correct typos and revise abbreviations according to the literature.
Round 2
Reviewer 2 Report
I have no further comments on the article.